# High-Throughput Microfluidic Production of Ultrasmall Lecithin Nanoliposomes for High-Efficacy Transdermal Delivery and Skin-Aging Treatment

**DOI:** 10.3390/biomedicines13020322

**Published:** 2025-01-30

**Authors:** Xiao Liang, Chan Lu, Fangqiao Zheng, Zhengyi Lan, Haoji Wang, Muhammad Shafiq, Xinxin Pan, Hangrong Chen, Ming Ma

**Affiliations:** 1School of Chemistry and Materials Science, Hangzhou Institute for Advanced Study, University of Chinese Academy of Sciences, Hangzhou 310024, China; liangxiao22@mails.ucas.ac.cn (X.L.); luchan23@mails.ucas.ac.cn (C.L.); zhengfangqiao23@mails.ucas.ac.cn (F.Z.); wanghaoji21@mails.ucas.ac.cn (H.W.); hrchen@mail.sic.ac.cn (H.C.); 2State Key Laboratory of High Performance Ceramics and Superfine Microstructures, Shanghai Institute of Ceramics, Chinese Academy of Sciences, Shanghai 200050, China; lanzhengyi@mail.sic.ac.cn; 3Innovation Center of NanoMedicine (iCONM), Kawasaki Institute of Industrial Promotion, Kawasaki-ku, Kawasaki 210-0821, Japan; shafiqdr786@yahoo.com; 4School of Biotechnology, East China University of Science and Technology, Shanghai 200237, China

**Keywords:** ultrasmall nanoliposomes, microfluidics, transdermal drug delivery, photoaging, skin repair, wound healing

## Abstract

Background: Liposome particles with smaller sizes could increase transdermal drug delivery efficacy for enhanced skin penetration. While microfluidic methods have enabled controlled liposome synthesis, achieving efficient production of ultrasmall nanoliposomes (NLP_US_) with a size smaller than 40 nm yet remains an unmet challenge. Methods: In this study, we employed a helical-blade-strengthened co-flow focusing (HBSCF) device to efficiently synthesize NLP_US_, which demonstrated superior skin permeation and retention. Results: Liposome formulation primarily contains unsaturated lecithin, which endows an unprecedented capacity to NLP_US_ to scavenge reactive oxygen species (ROS). Moreover, NLP_US_ can effectively encapsulate a broad spectrum of anti-aging agents, including coenzyme Q10 (CoQ10), while preserving its physical properties. In a photoaged skin model, topical application of CoQ10-loaded NLP_US_ (CoQ10@NLP_US_) inhibited ultraviolet B (UVB)-induced matrix metalloproteinase-1 (MMP-1) production, and promoted collagen type I (Col-I) synthesis in skin cells, thereby effectively rejuvenating the photoaged skin. Conclusions: This study presents a straightforward and efficient method for the production of NLP_US_, thereby offering a promising platform for transdermal delivery of diverse therapeutic agents.

## 1. Introduction

The skin is the largest organ in human body and is well-suited for the topical and systemic delivery of therapeutics [1]. While drugs intended for systemic absorption require transdermal delivery, therapeutics targeting specific skin dysfunctions should remain on the skin surface for a sufficient time period (e.g., ultraviolet protection, prebiotics) [2,3]. However, the stratum corneum (SC), the outermost layer of the skin, presents a major barrier for drug penetration. The SC consists of 10–20 layers of keratinocytes along with a hydrophobic lipoprotein matrix, which protects the body from external insults, such as bacterial microorganisms and toxic chemicals [4,5]. It is due to this natural protective mechanism of the skin that transdermal drug delivery is challenging, especially for the introduction of large molecules that cannot easily traverse through the SC, thereby limiting drug delivery efficacy [6,7]. To overcome the transdermal barrier, most of the time, high drug doses are required, albeit with side effects [8]. Recently, significant efforts have been made to enhance drug penetration through the SC by utilizing both active and passive methods [9]. Active methods, including electroporation [10,11], microneedling [12,13], and sonophoresis [14,15], often require additional equipment and they only temporarily improve drug penetration. In contrast, passive methods, such as liposomes [16,17] micro/nanoemulsions [18,19], and dendrimers [20,21] enable spatio-temporal drug delivery, thereby simplifying the process while maintaining permeation flux over time.

Liposomes, which are lipid-based vesicles composed of phospholipids and cholesterol, are widely employed as depots for drug delivery, thanks to their structural and morphological features similar to the lipid bilayer of the skin [22]. They can interact with hydrophobic components of the SC (e.g., lipids) to facilitate drug penetration [23]. In particular, liposomes with particle sizes in the range of 30–40 nm have been shown to significantly enhance skin permeability [24]. Nevertheless, the synthesis of ultrasmall nanoliposomes (NLP_US_) with a particle size below 40 nm yet remains an ongoing challenge.

Traditional liposome synthesis methods, such as lipid film hydration [25], reverse evaporation [26], and solvent injection [27], can produce large quantities of liposomes, albeit with an uncontrolled size and a heterogeneity in size distribution and structure. Alternative methodologies, such as membrane extrusion [25,27] or ultrasonication [28,29], can reduce liposome size, but they are time-consuming and may not ensure batch-to-batch consistency [25,26]. Moreover, synthetic surfactants, which are often used to facilitate liposome production, can accumulate in the body and cause adverse effects [30]. To the best of the authors’ knowledge, batch synthesis of NLP_US_ with uniform particle size is almost impossible with traditional macroscopic methods. The advent of microfluidic technology has addressed most of the limitations associated with traditional liposome synthesis, which is ascribed to the precise control over the particle size and dispersion [31,32,33]. However, several challenges remain for the large-scale synthesis of NLP_US_, partly owing to the inherent constrains of the microfluidic chip designs, thereby limiting the broader application of liposomal drugs.

In our previous report, we introduced an HBSCF device, which can produce nanoliposomes with a particle size of 80 nm and a low polydispersity index (PDI) [34]. Nevertheless, the utility of the HBSCF device for the production of NLP_US_ yet remains to be elucidated. In this work, we optimized the synthesis parameters, such as total flow rate (TFR, the sum of the internal and external phase flow rates) and flow rate ratio (FRR, external phase: internal phase), to produce lecithin-based nanoliposomes with particle sizes below 40 nm with an excellent homogeneity (PDI < 0.2), while obviating the use of the synthetic surfactants. The ability of NLP_US_ to load a variety of anti-aging agents, including coenzyme Q10 (CoQ10), without altering their particle size or homogeneity demonstrates the versatility of this system as a drug carrier (Figure 1). Furthermore, the unsaturated lipid nature of lecithin may endow the nanoliposomes with an ability to efficiently scavenge reactive oxygen species (ROS), thereby rendering them an effective transdermal drug delivery system for skin photoaging models [35]. In vitro transdermal experiments demonstrated that our NLP_US_ could deliver drugs more efficiently to the basal layer of the skin as compared to the conventional liposomes with a larger size (>200 nm, NLP_NM_). To verify the potential of the CoQ10-loaded NLP_US_ (CoQ10@NLP_US_) to repair photoaged skin, we used a mouse model of photoaging; the latter closely recapitulates the pathophysiological characteristics of aged human skin. The HBSCF device may offer a promising platform for the production of homogeneous NLP_US_ for enhanced transdermal drug delivery, which may also have broad implications for liposomal drug delivery for skin regeneration and potentially other related disciplines.

## 2. Materials and Methods

### 2.1. Materials, Cell Lines, and Animal Models

The lipid egg phosphatidylcholine (EPC) was purchased from Shenyang Tianfeng Bio-Pharm Co., Ltd. (Shenyang, China). Cholesterol (Chol) and dicetyl phosphate (DCP) were bought from Shanghai Rhawn Reagent Co., Ltd. (Shanghai, China). Coenzyme Q10 (CoQ10) was bought from Chenguang Biotech Group Co., Ltd. (Shijiazhuang, China).

Fluorescein isothiocyanate (FITC), 1,1′-dioctadecyl-3,3,3′,3′-tetramethylindocarbocyanine perchlorate (DiI), 2-(4-Amidinophenyl)-6-indolecarbamidine dihydrochloride (DAPI), and phosphate-buffered saline (PBS) were acquired from Beijing Leagene Biotech Co., Ltd. (Beijing, China).

The back skin of Bama miniature pigs (weight, 5~6 kg) was bought from Aperture Biotechnology (Shandong) Co. (Jinan, China). Human skin fibroblasts (HSFs) and Dulbecco’s Modified Eagle Medium (DMEM) were both sourced from Cellverse Co., Ltd. (Shanghai, China). Cell Counting Kit-8 (CCK-8) was bought from Shanghai Qihai Futai Biotechnology Co., Ltd. (Shanghai, China). Senescence β-Galactosidase Staining Kit and 2,7-dichlorodihydrofluorescein diacetate (DCFH-DA) were bought from Beyotime Biotechnology (Shanghai, China). Collagen type I (Col-I) immunohistochemistry kit and matrix metalloproteineases-1 (MMP-1) immunohistochemistry kit were purchased from Uscn Sciences Co., Ltd. (Wuhan, China). HSFs were cultured in DMEM supplemented with 1% penicillin-streptomycin, 1% fibroblast growth factor (FGF), and 15% fetal bovine serum (FBS) at 37 °C in a 5% CO_2_ atmosphere. Institute of Cancer Research (ICR) mice and nude mice (female, aged 6–8 weeks) were acquired from Shanghai Lab. Animal Research Center (Shanghai, China).

### 2.2. Preparation of Lipid Solution as an Internal Phase

A vacuum dryer was used to fabricate lipid films. An initial lipid mixture composed of EPC, Chol, and DCP was prepared in chloroform at a molar ratio of 5:4:1. Uncapped vials were placed in a vacuum oven at 40 °C for at least 12 h to evaporate the solvent. Since unsaturated lipids are prone to oxidation in the presence of moisture or oxygen, ethanol was dehydrated with 20 wt% molecular sieves for at least 24 h. The dried lipid mixture was then rehydrated with anhydrous ethanol. For drug-loaded formulations, therapeutics (5 wt% with respect to the weight of the lipid components) were loaded into initial lipid mixture.

### 2.3. Fabrication and Evaluation of Liposomes

Nanoliposomes were synthesized with HBSCF and normal co-flow focusing (CF) devices, separately. Both of the above devices have dimensions similar to the previously reported devices and were fabricated using a projection microstereolithography-based 3D printing method provided by BMF Precision Tech Inc. (BMF, Shenzhen, China) [34]. For liposome synthesis, poly (ethylene glycol) (PEG, Mw = 400 Da) was dissolved in PBS (0.01 mol/L, pH = 7.4) at a concentration of 15 mg/mL, which served as an external phase solution. The lipid solution was used as an internal phase solution. These solutions were simultaneously injected into the microfluidic device (either CF or HBSCF device) with a precision syringe pump for internal phase solution and an advection pump for the external phase solution. It is worth noting that all experiments were performed at an ambient temperature, ~25 °C, as a lower temperature can lead to the precipitation of the components before liposome formation. The particle size of liposomes was measured by a Dynamic Light Scattering (DLS) particle size analyzer (Malvern, Worcestershire, UK) at 25 °C. The samples stored in a refrigerator at 4 °C were brought to room temperature before analysis. Morphology of liposomes was investigated by cryo-transmission electron microscopy (Thermo Fisher Scientific, Waltham, MA, USA).

### 2.4. In Vitro Skin Permeation

The in vitro transdermal ability of NLP_US_ was studied with the Franz diffusion cell method. To ensure the reproducibility of skin penetration experiments, two parallel samples were evaluated for each group. The back skin of Bama miniature pigs (weight, 5~6 kg) was processed for the isolation of subcutaneous adipose tissues and fixed between the supply pool and the receiving pool; the SC faced the supply pool. To visualize transdermal infiltration, 1 mL of FITC-labelled NLP_US_ (synthesized at a higher TFR by HBSCF device) or FITC-labelled NLP_NM_ (synthesized at a lower TFR by CF device) were added to the supply pool and uniformly applied to the pig skin. During transdermal infiltration, the skin was removed at different time points (e.g., 10 min, 2 h, 4 h, etc.), and tissue sections were prepared with a cryosectioner (Leica, Wetzlar, Germany). The distribution of FITC in skin tissue sections was observed with a laser confocal microscope (Leica, Wetzlar, Germany).

Moreover, in vitro transdermal analysis of the CoQ10 was performed using Raman spectroscopy. Free CoQ10 (1 mL), CoQ10@NLP_US_ (1 mL), and CoQ10@NLP_NM_ (1 mL) containing similar concentrations of CoQ10 (1.78 μg/mL) were added to the supply pool and were uniformly applied to the pig skin. In vitro transdermal assays were performed at 300 rotations per minute (rpm) and 37 °C. Following transdermal experiments, skin samples were protected from the light for up to 1 h and were subjected to Raman detection with a Raman spectrometer (Renishaw, Wotton-under-Edge, UK).

### 2.5. Biocompatibility Assay

To delineate the feasibility of CoQ10@NLP_US_ to treat photoaged cells, the influence of materials on cell viability was first assessed. Cell viability was studied using the CCK-8 assay, and absorbance at 450 nm was measured with a microplate reader (Thermo Fisher Scientific, Waltham, MA, USA). HSFs were seeded in a 96-well plate at a density of 2 × 10^4^ cells per well, and after 24 h, the medium was aspirated and cells were washed with PBS. Thereafter, 100 µL of sample with different concentrations (i.e., CoQ10, NLP_US_, and CoQ10@NLP_US_) was dissolved in the medium and added into the wells for 24 h. Thereafter, cells were washed with PBS followed by the addition of 100 μL of PBS containing 10% (*v*/*v*) CCK-8 into each well. Cells were incubated for 1 h at 37 °C while being protected from light, and the absorbance of the supernatant was measured at 450 nm with a microplate reader.

### 2.6. UVB-Based Cellular Model of Photoaging

We further investigated optimal exposure time for modeling photoaged cells and used ultraviolet B (UVB) irradiation for different time points. HSFs were seeded at a density of 2 × 10^4^ cells per well in a 96-well plate. After incubation for up to 24 h, the medium was aspirated to prevent the effect of serum and other components of the medium on UVB uptake, and PBS was added to the cells. HSFs were irradiated with a narrow-spectrum UVB lamp (Philips, Eindhoven, NLD) at an irradiation intensity of 5 mW/cm^2^. Radiation exposure was regulated by adjusting the irradiation time. It is worth noting that the irradiation intensity was calibrated before each experiment with an irradiometer; the latter was purchased from Beijing Shida Photoelectric Technology Co., Ltd. (Beijing, China). Subsequently, PBS was aspirated, FBS-free DMEM was added, and cells were incubated for 24 h at 37 °C. Cell viability was studied with the CCK-8 assay.

### 2.7. Cell Uptake Assay

HSFs were seeded at a density of 2 × 10^5^ cells per well in 35 mm glass-bottomed confocal dishes and cultured for 24 h. The medium was aspirated and cells were washed with PBS one time. DiI@NLP_US_ or free DiI solution were added to petri dishes for 10, 20, and 30 min. The staining solution was removed and cells were washed with PBS and DAPI solution for 5 min. DAPI solution was removed and cells were washed once with PBS and observed with an inverted fluorescence microscope (Leica, Wetzlar, Germany). To prevent fluorescence quenching, the entire staining procedure was performed under light protection.

### 2.8. β-Galactosidase Assay

The β-galactosidase (β-gal) activity was detected by Senescence β-Galactosidase Staining Kit. After modeling cellular aging in a 6-well plate, cell culture medium was aspirated and cells were washed with PBS. A total 1 mL of β-gal fixative solution was added at r.t. and cells were fixed for up to 15 min. Cell fixative was aspirated and cells were washed with PBS three times for up to 3 min. PBS was aspirated, 1 mL of staining solution was added, and samples were incubated at 37 °C overnight. Finally, cells were observed with an inverted light microscope (Olympus, Tokyo, Japan).

### 2.9. Flow Cytometry and Analysis

Quantitative analysis of ROS was performed using a flow cytometer (BD Biosciences, San Jose, CA, USA). Initially, a photoaged cell model was established as described previously (see Section 2.6). Thereafter, UVB-irradiated HSFs were co-incubated along with CoQ10, NLP_US_, or CoQ10@NLP_US_ for up to 24 h. HSFs were stained according to the protocol provided by the DCFH-DA probe supplier, and repeatedly washed with PBS to minimize the residual drug. HSFs were then digested with ethylenediaminetetraacetic acid (EDTA)-free trypsin, washed with PBS, and finally analyzed by flow cytometry (BD Biosciences, San Jose, CA, USA).

### 2.10. Animal Model, Groups, and Treatment

A skin photoaging model was established using a UVB lamp (Philips, Eindhoven, The Netherland). Different types of samples (PBS, CoQ10, NLP_US_, and CoQ10@NLP_US_) with Aquaphor (Eucerin) were evenly spread on the backs of nude mice (female, 6 weeks old); the latter were placed in a dark environment for 2 h before UVB irradiation at an intensity of 300 mJ/cm^2^, a process that was continued for up to 7 days. Nude mice were randomly divided into 5 groups (*n* = 4): (i) no UVB irradiation, PBS cream (per day × 7 days), (ii) UVB irradiation, PBS cream (per day × 7 days), (iii) UVB irradiation, CoQ10 cream (per day × 7 days), (iv) UVB irradiation (per day × 7 days), NLP_US_ cream, and (v) UVB irradiation, CoQ10@NLP_US_ cream (per day × 7 days).

### 2.11. ELISA for Collagen and Matrix Metalloproteinases

Cell samples were successfully modeled by UVB irradiation and treated with various samples for up to 24 h (see Section 2.6). Skin samples were collected from photoaged skin areas of the backs of nude mice that had been irradiated with UVB for 7 days and treated with various groups. Subsequently, equal volumes of skin tissues were taken and the concentrations of Col-I and MMP-1 in explanted skin tissues were determined following manufacturer’s instructions.

### 2.12. Safety Evaluation

Safety was evaluated using ICR mice and the backs of the mice were shaved one week in advance to minimize the shaving effect on the skin. ICR mice were randomly divided into four groups (*n* = 5), and samples, including PBS, CoQ10, NLP_US_, and CoQ10@NLP_US_, were locally applied to the skin. All formulations were mixed with Aquaphor (Eucerin, DEU). After drug treatment, mice were placed in a dark environment for 2 h to avoid light-mediated photolysis. The appearance of the skin was recorded daily, while weight was measured every two days during the treatment. At day 14, mice were euthanized and major organs (e.g., heart, liver, spleen, lungs, kidneys, skin, etc.) as well as blood were collected for histological and hematological blood tests.

### 2.13. Ethical Statement

All animal studies of mice experiments were conducted with the protocols approved by the Institutional Animal Care and Use Committee of Shanghai Rat&Mouse Biotech Co., Ltd. (approval number: SHRM-IACUC-068).

## 3. Result and Discussion

### 3.1. Controllable Microfluidics for the Synthesis of NLP_US_

Microfluidics is a promising technique for the fabrication of nanoliposomes, which may enable precise control over the particle size as well as homogeneity of liposomes. We previously fabricated an HBSCF device, which enabled the production of nanoliposomes with lower PDI values in terms of the particle size at a high TFR (Appendix A) [34]. Herein, we developed a nanoliposome synthesis apparatus based on the HBSCF device (Figure 2A). Briefly, we used a high-speed advection pump as a power source for an external phase and a precision syringe pump for an internal phase, which can allow a flow rate higher than that of the 80 mL/min to the whole system during the fabrication process. During microfluidic synthesis with an HBSCF device, the lipid/ethanol mixture (inner phase) is rapidly diluted with PBS (outer phase) and the fragments of the lipid bilayer are self-assembled to form nanoliposomes. The dilution process was significantly increased with a specially designed helical vane structure in the flow channel of the HBSCF device. Additionally, we optimized the phospholipid membrane formation process and introduced PEG (Mw = 400 Da) into an external phase solution. The synergistic effect of these modifications led to the formation of NLP_US_. It is well-known that the lipid component is a critical component for the synthesis of liposomes, which influences physical properties, such as stability, size, and toxicity [36]. We used natural ingredients, including EPC, Chol, and DCP, to fabricate liposomes due to their negligible toxicity and good biocompatibility. Unlike traditional microfluidic devices, the size of liposomes prepared with an HBSCF device can be precisely controlled by varying the relative flow rates of the PBS and lipid/ethanol mixture.

The influence of the PBS: lipid FRR on the hydrodynamic diameter and PDI of liposomes is shown in Figure 2B,C. Herein, the TFR was 80 mL/min, the lipid concentration was 25 mM, and the FRR varied in the range of 10–100. As shown in Figure 2B, the hydrodynamic particle size of liposomes first decreased with an increase in the FRR, and then increased with further increases in the FRR. Below an FRR value of 40, the hydrodynamic particle size of liposomes rapidly decreased with an increase in the FRR; the hydrodynamic particle size of the liposomes was 45 nm at an FRR value of 40. With the subsequent increase of FRR, the particle size of liposomes slowly increased. It is worth noting that the particle sizes of liposomes synthesized with an HBSCF device were smaller than that of liposomes synthesized by a CF device at all FRR values. We further screened the effect of the TFR on the size of liposomes and observed a trend similar to the FRR (Appendix A). The PDI values of nanoliposomes were below 0.2 with narrow size distribution, which is indicative of the good homogeneity of the liposomes (Appendix A).

Cryo-transmission electron microscopy (Cryo-TEM) further revealed distinct advantage of the HBSCF device for NLP_US_ synthesis, with homogeneous particle sizes as compared to the CF device (Figure 2D,F). As shown in Figure 2E, the typical bilayer membrane structure of liposomes can be clearly observed, thereby affirming the successful synthesis of NLP_US_. We also quantified the particle sizes in cryo-TEM images using image J software (version 1.54g). The distribution of particle sizes of NLP_US_ prepared using an HBSCF device was mostly in the range of 15–35 nm (Figure 2G). In contrast, a proportion of particles with size larger than 45 nm were obtained by the CF device (Figure 2H), thereby demonstrating an inferior performance of the CF for the regulation of liposome size than that of the HBSCF device. The dispersion stability of NLP_US_ is also an important factor, which can play a pivotal role for the application of NLP_US_. We delineated variation in the particle size of the NLP_US_ for up to 2 days (Figure 2I). While NLP_US_ exhibited only a slight increase in the particle size over a period of 48 h, the particle size measured by DLS at 48 h was still less than 60 nm, which is indicative of good dispersion stability of the NLP_US_. Additionally, we observed the formation of a multilayer NLP at a smaller TFR value, which can be attributed to a higher untapped potential of the HBSCF device (Appendix A).

Since liposomes are an excellent drug carrier and our HBSCF device may enable the synthesis of NLP_US_ with particle size below 40 nm, we envision that NLP_US_ can effectively modulate the pharmacodynamic and pharmacokinetic properties of liposomal drugs. As a proof-of-concept, we prepared different types of drug-loaded liposomes with an HBSCF device by using fluorescein as well as model antioxidative therapeutics, including DiI and CoQ10. Figure 3A illustrates the schematic diagram of drug-loaded liposomes; therapeutics are mainly located within the phospholipid bilayer of the NLP_US_. For each type of model drug, we synthesized drug-loaded liposomes using both an HBSCF device and a CF device as well as delineated the effect of FRR on the particle size and PDI values. NLP_US_ with or without drug loading displayed homogenous particle sizes with PDI values below 0.2, which was further corroborated by DLS results (Figure 3B,C and Appendix A). Mirroring NLP_US_ with no drug loading, an increase in TFR at a fixed FRR initially increased the particle size, which was then stabilized with a further increase in the TRR (Appendix A). The morphology of drug-loaded NLP_US_ was discerned using Cryo-TEM. Figure 3D,E exhibits the Cryo-TEM images of liposomes loaded either with CoQ10 or DiI, respectively. The typical bilayered structure of liposomes was clearly observed, thereby indicating the successful synthesis of drug-loaded liposomes. Quantitative analysis with image J showed that the particle size of drug-loaded liposomes was below 40 nm (Figure 3F,G). The dispersion stability of the drug was not affected even after its encapsulation in NLP_US_, which may have implications for the applications of NLP_US_ (Appendix A). These results manifested the potential feasibility of the HBSCF device for the synthesis of drug-carrying NLP_US_.

### 3.2. Transdermal Mechanisms of NLP_US_

For non-invasive transdermal drug delivery, we synthesized FITC-labelled NLP_US_ with an HBSCF device and used a standard Franz diffusion cell system to preliminarily assess SC penetrability (Figure 4A). Additionally, we evaluated the penetrability of FITC-labelled NLP_US_ in the skin and measured fluorescence distribution. As shown in Figure 4B, the green fluorescence at 10 min was mainly distributed on the SC, hair follicles, and sebaceous glands. In contrast, the fluorescence of the FITC-labelled NLP_US_ group was appreciably stronger than that of the FITC-labelled NLP_NM_ (synthesized by the CF device) group. Further detailed evaluation of transdermal administration revealed that while the fluorescence of the FITC-labelled NLP_US_ group was mainly accumulated in the basal layer of the skin by 2 h post-administration, the FITC-labelled NLP_NM_ group still displayed fluorescence only on the SC. Quantitative analysis of fluorescence in skin sections using LAS X Office software (version 1.4.27713.5) revealed that while the fluorescence of the FITC-labelled NLP_US_ group attained its maxima in the SC as soon as 10 min, with negligible increase thereafter, fluorescence intensity in the basal layer was increased with an increase in the administration time. On the other hand, for the FITC-labelled NLP_NM_ group, fluorescence intensities were increased in a time-dependent manner for both SC and basal layer. It is worth noting that while the FITC-labelled NLP_US_ group attained maximum fluorescence intensity on the SC as earlier as 10 min, the FITC-labelled NLP_NM_ group attained an equivalent fluorescence intensity by 2 h (Figure 4C,D).

Since CoQ10 lacks an intrinsic fluorescence, confocal Raman spectroscopy was used to further delineate the transdermal penetrability of the NLP_US_ for CoQ10. Raman spectroscopy detects characteristic vibrational energy levels of molecules irradiated by a laser beam and provides information about the molecular structure of tissue components while obviating fluorescent labeling or chemical staining [37,38]. Consequently, confocal Raman spectroscopy may help discern changes in the structure of the skin components alongside tracking the penetration of exogenous substances [37]. For preliminary assessment of the characteristic peaks for CoQ10 permeation, we collected Raman spectra of both skin as and free CoQ10 (Figure 4E). While CoQ10 exhibited three strong characteristic peaks in the range of 750–880 cm^−1^, the skin lacked characteristic peaks in this region, thereby suggesting that these peaks can be used as markers to elucidate the penetration of CoQ10 in the skin. The peaks at 750 cm^−1^ and 880 cm^−1^ are attributed to the ring torsional deformation as well as ring bend deformation of the quinone moieties of the CoQ10, while the peak at 800 cm^−1^ is ascribed to the carbonyl groups of the quinone moieties of the CoQ10 [38]. Figure 4F–H shows the Raman spectra of the skin surface, 20 μm from the skin surface, and 60 μm from the skin surface treated with CoQ10@NLP_US_ solution, CoQ10@NLP_NM_ solution, and free CoQ10 solution. While both CoQ10@NLP_US_ and CoQ10@NLP_NM_ groups exhibited characteristic peaks of CoQ10 on the skin surface, the CoQ10@NLP_US_ group exhibited stronger peak intensity than that of the CoQ10@NLP_NM_ group. In contrast, the skin surface treated with free CoQ10 did not exhibit the above-mentioned characteristics peaks of CoQ10. With a further increase in the scanning depth from the skin surface (e.g., 20 μm and 60 μm beneath the skin), only the CoQ10@NLP_US_ group displayed characteristic peaks of CoQ10, while the remaining two groups had no characteristic peaks appearing anymore.

These data showed that the NLP_US_ can effectively load therapeutics, enhance drug penetration through the SC barrier of the skin, and increase the residence time of the drug in the skin, which may have broad implications for the treatment of various dermatological disorders while avoiding the risks associated with the systematic administration of therapeutics.

### 3.3. Repair of Photoaged HSFs by CoQ10@NLP_US_

To validate pharmacokinetic ability of liposomal drugs synthesized with an HBSCF device, we first performed cell experiments with liposomal drugs in vitro. To observe cellular uptake, DiI@NLP_US_ was prepared with an HBSCF device and incubated along with HSFs for cellular uptake by confocal microscopy. HSFs incubated along with DiI@NLP_US_ and DiI manifested fluorescence as soon as 10 min; fluorescence intensity was increased with an increase in the incubation time as evaluated for up to 30 min (Appendix A). Moreover, the fluorescence of cells incubated with the DiI@NLP_US_ group was mostly distributed on the cell membrane, while for DiI-treated cells, fluorescence intensity was uniformly distributed on the whole cell membrane as evaluated in enlarged images 10 min post-incubation (Appendix A). These data indicated that DiI@NLP_US_ penetrated HSFs via cellular uptake, rather than dye-mediated cellular disruption. Overall, NLP_US_ could deliver encapsulated drug into the cell’s interior in a very short time. The cytotoxicity of CoQ10@NLP_US_ was assessed in HSFs. As shown in Appendix A, NLP_US_, CoQ10, and CoQ10@NLP_US_ exhibited negligible cytotoxicity as evaluated with varying concentrations of the CoQ10 (0–60 μg/mL). Consequently, 60 μg/mL of CoQ10 was used for the subsequent experiments.

UVB rays are widely utilized to induce cellular senescence. We discerned cytotoxic effects of UVB with varying intensities on cells. The viability of HSFs was decreased in an intensity-dependent manner of the UVB (Figure 5D). We next chose a UVB intensity of 150 mJ/cm^2^ for cell irradiation. It is worth noting that this particular dose was chosen based on the photoaging of the cells, albeit with 80% cell viability.

We next deciphered the effect of CoQ10@NLP_US_ on UVB-induced photoaged cells (see Figure 5A for the experimental procedure illustrating the cellular photoaging with UVB). NLP_US_, CoQ10, and CoQ10@NLP_US_ mitigated UVB-induced photoaging of HSFs (Figure 5B). While the number of β-gal+ senescent cells (stained in green) decreased in all groups, CoQ10@NLP_US_ outperformed other groups and exhibited the lowest number of β-gal^+^ senescent cells, thereby indicating its beneficial effect in alleviating UVB-mediated cell photoaging. It has been previously reported that sustained exposure to UVB induces ROS in cells, which triggers cellular photoaging [39,40]. To investigate the mechanism by which CoQ10@NLP_US_ can alleviate cellular photoaging, we evaluated ROS levels in the photoaged cells treated with blank NLP_US_, CoQ10, and CoQ10@NLP_US_ with fluorescent staining and flow cytometry (Figure 5C,E and Appendix A). ROS levels were significantly increased in the HSFs with UVB irradiation; CoQ10@NLP_US_ exhibited a significantly higher ROS scavenging ability than that of CoQ10, and manifested ROS levels similar to unirradiated cells. The ROS scavenging ability of CoQ10@NLP_US_ can be ascribed to higher cellular uptake of CoQ10. NLP_US_ also exhibited an excellent ROS scavenging ability partly due to the EPC; the latter is an unsaturated phospholipid and is the main constituent of NLP_US_, and can scavenge intracellular ROS. Consequently, CoQ10 and NLP_US_ can show a combined effect to scavenge ROS. These data showed that UVB-induced cellular photoaging can be efficiently alleviated by CoQ10@NLP_US_, which is suggestive of its ability to restore redox homeostasis in HSFs.

UVB-induced oxidative stress can increase the expression of MMP-1 and induce the degradation of Col-I, thereby mirroring the cellular photoaging process [41,42]. The CoQ10@NLP_US_ significantly reduced the MMP-1 level and increased the Col-I level (Figure 5F,G) more than that of the CoQ10, which is indicative of the superior anti-aging effect against UVB-mediated increase in the MMP-1 levels. Taken together, CoQ10@NLP_US_ effectively scavenged UVB-induced ROS and alleviated the symptoms of photoaging in HSFs.

### 3.4. Topically Applied CoQ10@NLP_US_ to Repair Photoaged Skin

In vitro transdermal experiments demonstrated that NLP_US_ possessed excellent skin penetration and retention, suggesting its ability to deliver drugs into the basal layer of the skin and exert sustained therapeutic action. Given the excellent transdermal delivery efficiency of NLP_US_ and the reliable repair capacity observed in HSFs, we further evaluated the therapeutic potential of CoQ10@NLP_US_ for the treatment of mice photoaged skin model in vivo. Multiple skin irritation tests were performed on mice prior to the experiments, and results demonstrated the safety of CoQ10, NLP_US_, and CoQ10@NLP_US_ (Appendix A). Next, nude mice were randomly divided into five groups: control group (without UVB treatment) and four groups exposed to UVB (300 mJ/cm^2^) and treated with PBS, CoQ10, NLP_US_, and CoQ10@NLP_US_, respectively (100 μL solution containing 60 μg/mL CoQ10) on a daily basis (Figure 6A). The body weights of the nude mice remained stable throughout the experimental cycle (Figure 6E). Skin conditions on the backs of nude mice were recorded over a period of one week, with special attention given to the typical clinical signs of photoaged skin, including erythema, edema, and crusting damage on the skin surface. At week 1, PBS and CoQ10 groups exhibited severe scabs and erythema on the dorsal skin, whereas the NLP_US_ and CoQ10@NLP_US_ groups displayed different degrees of improvement (Figure 6B). These findings indicated that NLP_US_ possessed a good ability for photoaged skin repair. At day 7 after the treatment, the dorsal skin was collected for histological analysis. H&E staining manifested an appreciable increase in epidermal thickness within the UVB group, which was accompanied with the infiltration of inflammatory cells as compared to the control group (Figure 6C). These changes represented typical skin aging symptoms [43]. The epidermal thickness in the CoQ10 group was similar to that of the UVB group, thereby indicating that CoQ10 alone had difficulty penetrating into the SC barrier to exert its therapeutic. In contrast, NLP_US_ and the CoQ10@NLP_US_ successfully alleviated photoaging symptoms; CoQ10@NLPUS outperformed the NLP_US_ in terms of the treatment efficacy (Figure 6C).

The external manifestation of skin photoaging is the formation of wrinkles, which is ascribed to a reduction in the collagen content. Consequently, we further stained the tissue sections with Masson’s trichrome (MT) staining to visualize collagen fibers. Unlike the control group, which manifested intact and well-organized collagen fibers, the model as well as the CoQ10 groups exhibited disorganized collagen fibers. The NLP_US_ and CoQ10@NLP_US_ groups showed collagenous tissues similar to the control group (Figure 6D). To further delineate the level of collagen synthesis, Col-I and MMP-1 contents were detected by ELISA. While the UVB and CoQ10 groups exhibited significantly lower content of Col-I than that of the control group, the collagen content was found to be increased in the NLP_US_ and CoQ10@NLP_US_ groups (Figure 6F). On the other hand, the content of MMP-1, an enzyme responsible for the degradation of collagen, was significantly elevated both in the model and CoQ10 groups, while substantially reduced in the NLP_US_ and CoQ10@NLP_US_ groups (Figure 6G). It is worth noting that the Col-I and MMP-1 contents were comparable with the sham group in the CoQ10@NLP_US_ group (Figure 6F,G).

These results indicated that NLP_US_ possesses an intrinsic capacity to repair skin photoaging while simultaneously facilitating the penetration of the drug into the skin’s SC, and sustained release within the skin, thereby achieving the purpose of effectively alleviating UVB-induced skin photoaging.

## 4. Conclusions

In conclusion, this study employed the specially designed HBSCF device as a micro-mixer to produce NLP_US_ with a particle size of less than 40 nm. As a drug delivery system, NLP_US_ demonstrated the ability to efficiently load a broad range of therapeutics, maintaining a consistent particle size. They showed outstanding transdermal drug delivery efficacy in vitro as well as in a skin photoaging model. NLP_US_ efficiently penetrated the SC, thereby improving drug absorption through the skin and enhancing drug retention at the target site. Although the TFR of the synthesized NLP_US_ process exceeded 80 mL/min, it still remains insufficient as compared to the requirements of actual industrial production processes. Additionally, issues such as the differences between mouse skin and human skin, and the timing of safety evaluations, have not been adequately considered. All of the aforementioned issues require further investigation in subsequent studies. Nonetheless, this research contributes to advancing transdermal delivery of drugs and offers a method to minimize the toxic side effects associated with overdosing or the use of synthetic surfactants, while enhancing the therapeutic efficacy of the drugs. It also opens avenues for the industrial-scale application of liposomal drug formulations.

## Figures and Tables

**Figure 1 biomedicines-13-00322-f001:**
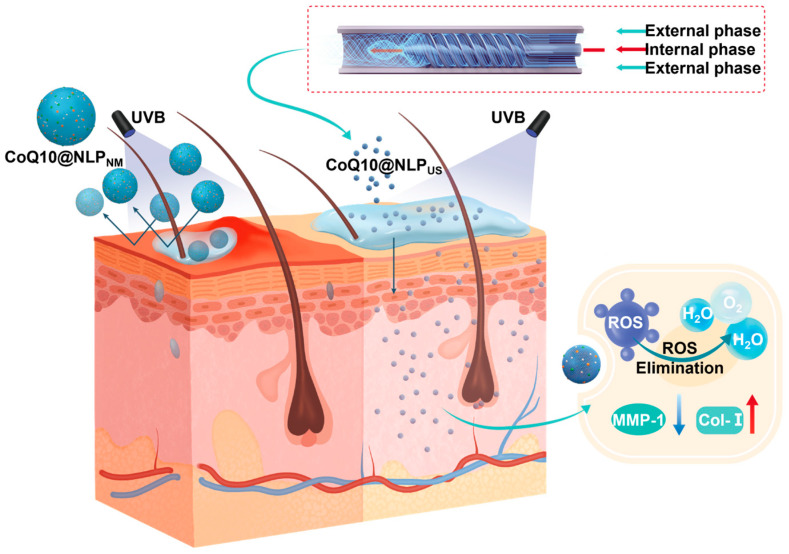
Schematic diagram illustrating the synthesis of CoQ10@NLP_US_ using the HBSCF device and its application as a non-invasive, highly efficient transdermal drug delivery system for the treatment of photoaged skin.

**Figure 2 biomedicines-13-00322-f002:**
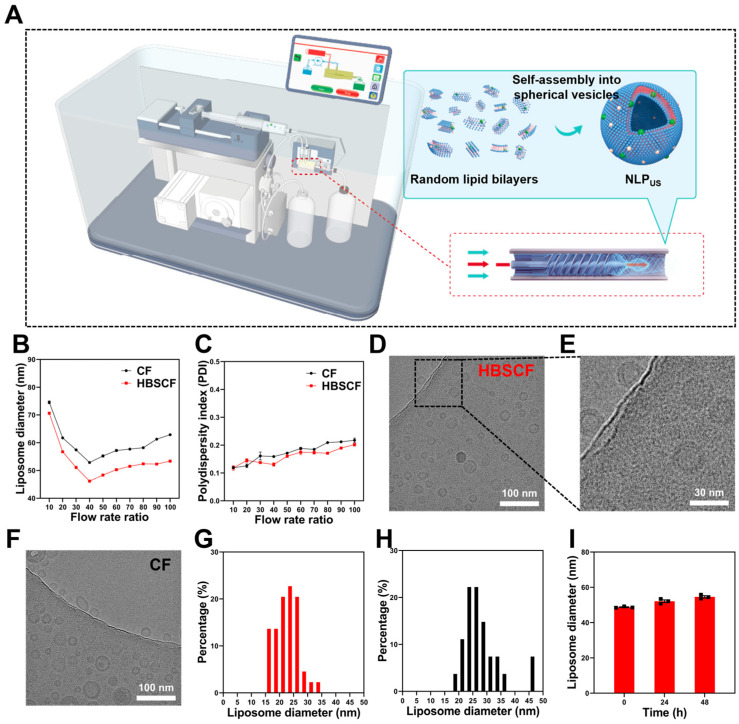
Preparation and characterization of NLP_US_. (**A**) Schematic representation of the procedure for the synthesis of liposomes. (**B**) DLS particle size of the nanoliposomes obtained at different FRRs and a fixed TFR of 80 mL/min. (**C**) PDI values of the nanoliposomes obtained at different FRRs and a fixed TFR of 80 mL/min. (**D**) Cryo-TEM image of NLP_US_ synthesized using HBSCF when TFR was fixed at 80 mL/min and FRR was fixed at 40 (scale bar = 100 nm). (**E**) Localized enlarged image of Figure 2D (scale bar = 30 nm). (**F**) Cryo-TEM image of NLP_US_ prepared with a CF device at a TFR value of 80 mL/min and an FRR value of 40 (scale bar = 100 nm). (**G**) The statistics of NLP_US_ particle sizes in Figure 2D. (**H**) The statistics of NLP_US_ particle sizes in Figure 2F. (**I**) Variation in the particle size of the NLP_US_ prepared with an HBSCF device over a period of 48 h. The values are represented as the mean ± SEM.

**Figure 3 biomedicines-13-00322-f003:**
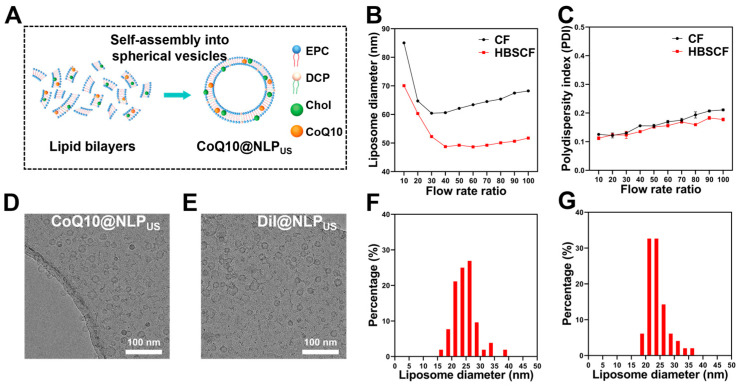
Characterization of NLP_US_ as a drug carrier. (**A**) Schematic illustration of drug-loaded nanoliposomes. (**B**) DLS particle size of the nanoliposomes containing CoQ10 obtained at different FRRs and a fixed TFR of 80 mL/min. (**C**) PDI values of the nanoliposomes containing CoQ10 obtained at different FRRs and a fixed TFR of 80 mL/min. (**D**) Cryo-TEM image of CoQ10@NLP_US_ prepared with an HBSCF device at a fixed TFR value of 80 mL/min and am FRR value of 40 (scale bar = 100 nm). (**E**) Cryo-TEM image of DiI@NLP_US_ prepared with an HBSCF device at a fixed TFR value of 80 mL/min and an FRR value of 40 (scale bar = 100 nm). (**F**) Distribution of the particle size of the CoQ10@NLP_US_. (**G**) Distribution of the particle size of the DiI@NLP_US_. The values are represented as the mean ± SEM.

**Figure 4 biomedicines-13-00322-f004:**
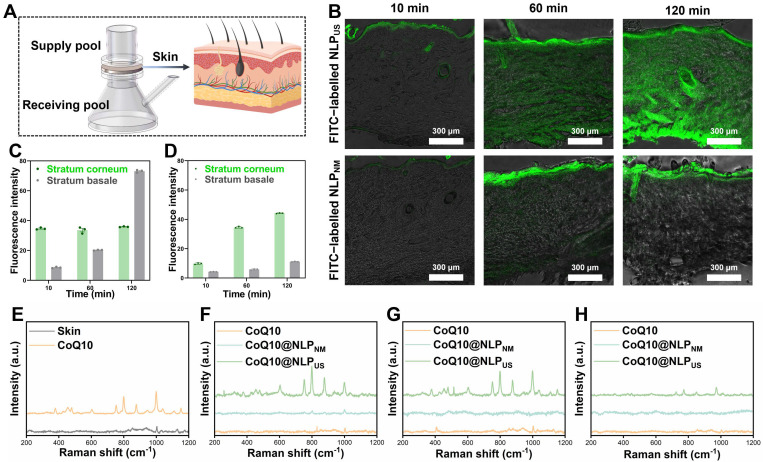
Transdermal characterization of NLP_US_. (**A**) Schematic illustration of the transdermal experiment. (**B**) Skin slices after transdermal experiments using FITC@NLP_US_ and FITC@NLP_NM_ (scale bar = 300 μm). (**C**) Fluorescence measurements of the stratum corneum and stratum basal in the FITC@NLP_US_ group of slices. (**D**) Fluorescence measurements of the stratum corneum and stratum basal in the FITC@NLP_NM_ group of slices. (**E**) Raman spectra of skin and CoQ10. (**F**) Raman spectra of the skin surface after performing transdermal experiments. (**G**) Raman spectra at 20 µm below the skin surface after performing transdermal experiments. (**H**) Raman spectra at 60 µm below the skin surface after performing transdermal experiments. The value is represented as the mean ± SEM.

**Figure 5 biomedicines-13-00322-f005:**
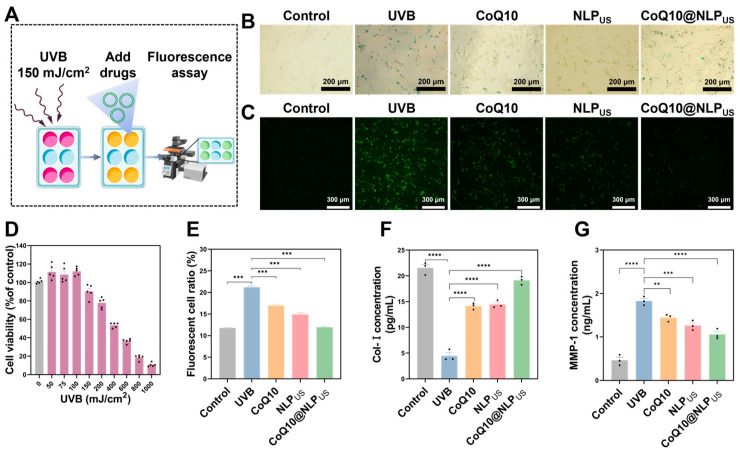
Repair of Photoaged HSFs by CoQ10@NLP_US_. (**A**) Schematic illustration showing cellular photoaging with UVB and its alleviation with the therapeutics. (**B**) Optical images of HSFs stained with β-gal (scale bar = 200 μm). (**C**) Representative ROS staining of HSFs after UVB exposure (150 mJ/cm^2^) and cocultured with different samples with a DCFH-DA probe (scale bar = 300 μm). (**D**) Effect of UVB intensities (50, 75, 100, 150, 200, 400, 600, 800, and 1000 mJ/cm^2^) on cell vitality after an exposure for up to 24 h. Control: non-irradiated cells. (**E**) The representative flow cytometry of the fluorescent cells after ROS staining. (**F**) Levels of Col-I in cell media as elucidated with ELISA (*n* = 3). (**G**) Levels of MMP-1 in cell media as measured with ELISA (*n* = 3). The values are represented as the mean ± SEM, and statistical significance was calculated via one-way ANOVA in GraphPad Prism (version 8.2004.3686). ** *p* < 0.01; *** *p* < 0.001; **** *p* < 0.0001.

**Figure 6 biomedicines-13-00322-f006:**
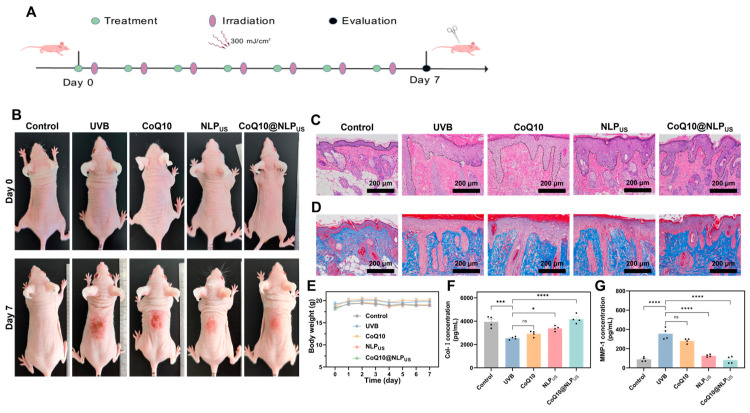
Topically applied CoQ10@NLP_US_ to repair photoaged skin in mice. (**A**) Schematic diagram of the construction of a mice skin photoaging model and CoQ10@NLP_US_ treatment. (**B**) Representative images of mouse skin before and after UVB damage. (**C**) Representative H&E staining of the epidermis, dermis, and subcutis for all mouse groups (scale bar = 200 μm). (**D**) Representative Masson trichrome collagen staining of the epidermis, dermis, and subcutis for all mouse groups. Collagen is blue (scale bar = 200 μm). (**E**) Body weights of mice during the experiment. (**F**) Levels of Col-I in photoaged skin tissues measured by ELISA (*n* = 3). (**G**) Levels of MMP-1 in photoaged skin tissues measured by ELISA (*n* = 3). The value is represented as the mean ± SEM, and statistical significance was calculated via one-way ANOVA in GraphPad Prism. * *p* < 0.05; *** *p* < 0.001; **** *p* < 0.0001; ns: no significance.

## Data Availability

The original contributions presented in this study are included in the article/Appendix A. Further inquiries can be directed to the corresponding authors.

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
