# Peer review of "High-Throughput Microfluidic Production of Ultrasmall Lecithin Nanoliposomes for High-Efficacy Transdermal Delivery and Skin-Aging Treatment"

_biomedicines, 2025, doi:10.3390/biomedicines13020322_

Round 1
Reviewer 1 Report
Comments and Suggestions for Authors
The comments and suggestions for authors are in the files attached to this form.

I strongly suggest that the authors have a native English colleague revise the article, as several grammatical errors are present. The article "the" was unnecessarily added before many words and should be removed. The article "the", was added before several words, and should be deleted.
Author Response
Please see the attachment. At the same time, we have made corrections to the obvious grammatical errors in this article.

Reviewer 2 Report
Comments and Suggestions for Authors
This manuscript describes a method for producing ultrasmall nanoliposomes (NLPUS) using a helical-blade-strengthened co-flow focusing (HBSCF) microfluidic device for enhanced transdermal drug delivery and skin-aging treatment. The study addresses the challenge of producing NLPUS (<40 nm) for improved transdermal drug delivery. The use of lecithin, cholesterol, and DCP for liposome formation offers advantages in terms of safety and biocompatibility. However, a more detailed mechanism by which the HBSCF device facilitates the formation of NLPUS compared to traditional methods is required. The manuscript needs to include a longer-term safety evaluation than a 14-day study on mice. The in vitro data on skin permeation is promising, but further discussion on translating these findings to in vivo efficacy is necessary. The novelty of using liposomes for transdermal drug delivery is not adequately justified. The manuscript could highlight the unique features of the HBSCF device and NLPUS compared to existing technologies. The additional figure is necessary to summarize the overall approach, from NLPUS production to drug delivery and potential therapeutic effects. The authors also need to address the limitations of microfluidic approach, such as scalability and cost-effectiveness for large-scale production.
Round 2
Reviewer 2 Report
Comments and Suggestions for Authors
The raised queries in the manuscript are adequately updated.